# Performance evaluation of Baermann techniques: The quest for developing a microscopy reference standard for the diagnosis of *Strongyloides stercoralis*

**Woyneshet Gelaye**[1]*, **Nana Aba Williams**[2], **Stella Kepha**[3], **Augusto Messa Junior**[4], **Pedro Emanuel Fleitas**[5], **Helena Marti-Soler**[2], **Destaw Damtie**[6], **Sissay Menkir**[6], **Alejandro J. Krolewiecki**[2,5], **Lisette van Lieshout**[7], **Wendemagegn Enbiale**[1], on behalf of the Stopping Transmission of Intestinal Parasites (STOP) project consortium¶

**1** Bahir Dar University, College of Medicine and Health Science, Bahir Dar, Ethiopia, **2** Barcelona Institute for Global Health (ISGlobal), Hospital Clínic—Universitat de Barcelona, Spain, **3** Kenya Medical Research Institution, Nairobi, Kenya, **4** Centro de Investigaçao em Saúde da Manhiça (CISM), Maputo, Mozambique, **5** Universidad Nacional de Salta, Instituto de Investigaciones de Enfermedades Tropicales/CONICET, Oran, Salta, Argentina, **6** Bahir Dar University, College of Science, Department of Biology, Bahir Dar, Ethiopia, **7** Department of Parasitology, Centre of Infectious Diseases, Leiden University Medical Centre (LUMC), Leiden, The Netherlands

¶ Membership of the Stopping Transmission of Intestinal Parasites (STOP) project consortium is provided in the acknowledgments
* woyn1978mam@yahoo.com

**Data Availability Statement:** All relevant data are within the manuscript and its Supporting Information files.

## Abstract

### Background

Soil-transmitted helminths (STH) are common in low and middle income countries where there is lack of access to clean water and sanitation. Effective diagnosis and treatment are essential for the control of STH infections. However, among STH parasites, *Strongyloides stercoralis* is the most neglected species, both in diagnostics and control strategies. Diagnostic methods cover different approaches, each with different sensitivities and specificities, such as serology, molecular techniques and microscopy based techniques. Of the later, the Baermann technique is the most commonly used procedure. In the literature, several ways have been described to perform the Baermann method, which illustrates the overall lack of a '(gold) reference standard' method for the diagnosis of *S. stercoralis* infection. In this study we have evaluated the performance of three Baermann techniques in order to improve the reference standard for the microscopic diagnosis of *S. stercoralis* infection thereby facilitating individual case detection, mapping of the disease and proper evaluation of treatment responses.

### Methods/Principal findings

A community based cross sectional study was conducted at Zenzelima, Bahir Dar Zuria Ethiopia. A total of 437 stool samples were collected and analyzed by the following procedures: conventional Baermann (CB), modified Baermann (MB), and modified Baermann

**Funding:** This project was funded by the EDCTP2 program supported by the European Union (grant number RIA2017NCT-1845 -STOP; www. stoptheworm.org) Horizon 2020 European Union Funding for Research and Innovation as part of ALIVE Study. SK is supported by THRiVE-2, a DELTAS Africa grant # DEL-15-011 from Wellcome Trust grant # 107742/Z/15/Z and the UK government. ISGlobal is supported by the Spanish Ministry of Science, Innovation and Universities through the "Centro de Excelencia Severo Ochoa 2019-2023" Program (CEX2018-000806-S), and support from the Generalitat de Catalunya through the CERCA Program. The funders had no role in study design, data collection and analysis, decision to publish, or preparation of the manuscript.

**Competing interests:** The authors have declared that no competing interests exist.

with charcoal pre-incubation (MBCI). The diagnostic sensitivity and Negative Predictive Value (NPV) of each technique was calculated using the combination of all the three techniques as a composite reference standard. Our result indicated that larvae of *S. stercoralis* were detected in 151 (34.6%) stool samples. The prevalence of *S. stercoralis* infection based on the three diagnostic methods was 9.6%, 8.0%, and 31.3% by CB, MB, and MBCI respectively. The sensitivity and NPV for CB, MB, and MBCI were 26.7% and 70.8%, 22.1% and 69.6%, and 87.0% and 93.2%, respectively. The MBCI showed significant difference (P- value = <0.001) in the sensitivity and NPV values when compared with CB and MB values. The agreement between CB, MB, and MBCI with the composite reference standard was 31.8%, 26.7%, 89.6%, respectively.

## Conclusion/Significance

Our results suggest the superior performance of MBCI. It is relatively easy to implement, simple to perform and comparatively cheaper. The CB is by far the commonly used method in routine diagnostic although this technique significantly underestimates the true burden of the disease and thereby contributing to the exclusion of *S. stercoralis* from the control strategies. Therefore, MBCI is recommended as a routine microscopy-based diagnostic test for *S. stercoralis* infection, particularly in settings where molecular procedures are not available.

### Author summary

Strongyloidiasis is a poverty-related neglected tropical disease which can cause serious and potentially life-threatening symptoms, in particular in immunocompromised hosts. *S. stercoralis* is not yet included in the strategies coordinated by WHO for the control of STH, but there are plans for the establishment of a control strategy by 2030. Therefore, diagnostics and control tools to implement that strategy are needed. Different diagnostic approaches are used in different parts of the world and there is no standard diagnostic approach which can be used for routine diagnostic services and field studies. In this study, 437 stool samples from Northwest Ethiopia were analyzed using conventional Baermann, modified Baermann and modified Baermann with charcoal pre-incubation techniques. Using these procedures, we found high prevalence of *S. stercoralis* infection in the study area. The modified Baermann with charcoal pre-incubation technique worked significantly better than the others in recovering the *S. stercoralis* larvae, while the conventional Baermann, the most used in routine diagnostics, underestimates the true burden of the disease. The key findings in this study are important for future planning of intervention and control strategies against strongyloidiasis.

## Introduction

*S. stercoralis* is a helminthic parasite affecting an estimated 386 million people worldwide, with the true burden of disease still unknown [1]. The prevalence of *S. stercoralis* infections reaches significant levels in some tropical and subtropical settings [2–4], but the global prevalence is believed to be underestimated due to the low sensitivity of the diagnostic tools [5–9]. Clinical and laboratory diagnostics play a critical role in the understanding of the epidemiology of the

disease, guiding the deployment of resources and the implementation and evaluation of intervention strategies. Currently active strategies for the control of STH under the guidance of the World Health Organization (WHO) do not include *S. stercoralis* as part of the strategy, but the establishment of a new list of targets for STH control programs by the WHO includes an efficient strongyloidiasis control program in school age children (SAC) by 2030 [10]; and this is further supported by the inclusion of ivermectin in WHO's list of essential drugs for the treatment of STH [11]. Key components of this strategy are diagnostic tools for adequate surveys and the inclusion of drugs active against *S. stercoralis*, such as ivermectin and moxidectin, in Mass Drug Administration (MDA) programs.

Diagnosis of *S. stercoralis* infection commonly relies on the detection of larvae from stool samples, tissue biopsies, and other clinical specimens such as bronchoalveolar lavage [12]. For the examination of stool samples, concentration procedures are essential as the number of excreted larvae is usually very low. Commonly used techniques are the Baermann technique and stool culture, for which either plain agar plates or a charcoal copro-culture procedure can be used. Stool cultures have a relatively high sensitivity, since they allow the parasite to enter the free living cycle, but the procedure is cumbersome and time-consuming [13]. On the other hand, serological and molecular techniques have been used as alternative diagnostic approaches for case detection in both endemic and non-endemic settings [14–18]. Polymerase Chain Reaction (PCR) based methods showed higher sensitivity than the conventional Baermann (CB) and copro-culture methods [2,4] however, they are expensive to implement as routine diagnostic tools specially in resource limited countries. Suboptimal sensitivity of microscopy-based techniques for the diagnosis of *S. stercoralis* infections has a negative impact on prevalence measurements and burden of disease estimations [19].

Among the microscopy-based techniques for the detection of *Strongyloides* larvae, the Baermann technique is the most often used in field studies and clinical trials; we have however found different Baermann technique approaches being used in different laboratories. To our knowledge, a proper comparison between different Baermann techniques has not been done, while there is a clear need for an inexpensive and suitable microscopy-based diagnostic method sensitive enough for use in large-scale field-based epidemiological studies and clinical trials. In this context we have conducted a study to compare the diagnostic performance of three Baermann techniques.

## Methods

### Ethics statement

The study protocol was approved by the Institutional Review Board of College of Medicine and Health Sciences at Bahir Dar University with reference number CHMS/IRB 03–008. Permission was also obtained from Amhara Public Health Institute, Bahir Dar Administration Health office and Zenzelima Health Center. Written informed consent was obtained from participants, parents or guardians for children who were under the age of 18 years. Additionally, verbal assent was obtained from children between ages of 10–17. All information was kept confidential and each participant with a positive result received appropriate treatment free of charge.

### Study setting

A community based cross sectional study was conducted from August 12 to August 31 2019 in Zenzelima, Bahir Dar Zuria, Bahir Dar Ethiopia (Fig 1C). *Bahir Dar* is the capital city of the Amhara National Regional State and *located* at the exit of the Blue Nile River from Lake Tana at an altitude of 1,820 m (5,970 ft) above sea level. The city is *located* approximately 578 km

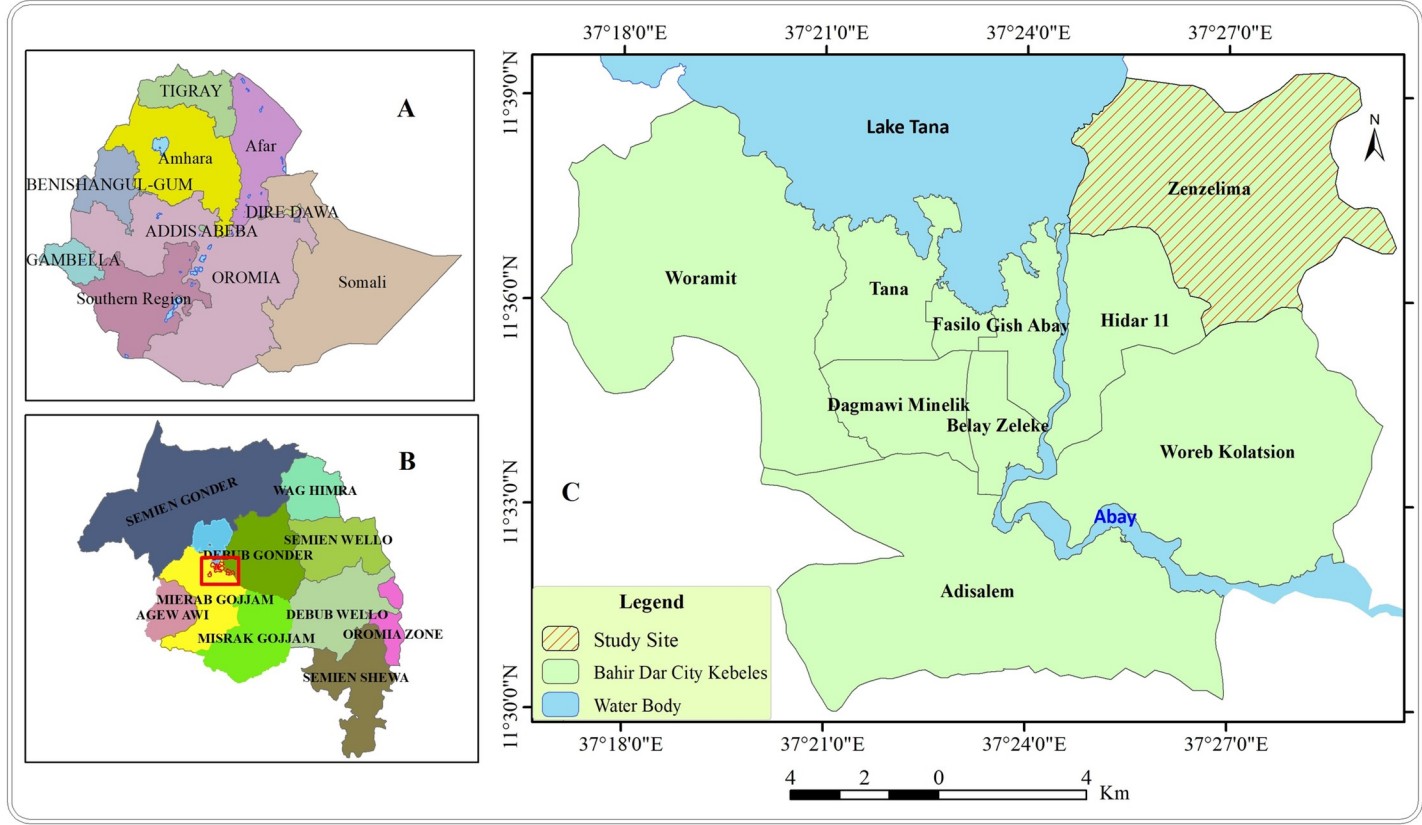

**Fig 1.** Location of the study site, Zenzelima: (A) Map of Ethiopia, (B) Map of Amhara region, (C) Zenzelima.

north-northwest of Addis Ababa, the country's capital. Zenzelima is situated about 7km north-east from Bahir Dar city (Fig 1C) with a total population of 11,203 with a distribution of 51% females (Health post coordinator, personal communication, October, 2019). The study was conducted in seven villages of Zenzelima kebele: Baynesa, Gedro, Gudguad, Kazimosh, Sesaberet -1, Sesaberet -2 and Zenzelima.

## Study population and sample size

With the assumption of 90% anticipated sensitivity ($S_N$), prevalence of 33% [6], precision of 0.05 (L) and 95% confidence interval ($Z^2_{\alpha/2}$), the minimum sample size required was calculated as 420 using Buderer's formula [20].

$$n = \frac{Z^2_{\alpha/2} \text{ x } S_N \text{ x } (1 - S_N)}{L^2 \text{ x } Prevalence}$$

All apparently healthy community members who live permanently in the study area and consented for the study were included in the study using simple random sampling. Participants who had taken anthelminthic drugs two months prior to the study period were excluded (S1 STARD–2015–Checklist).

## Data collection

**Socio-demographic characteristics.** Age, sex, and residence of the study participants were collected using standardized study form.

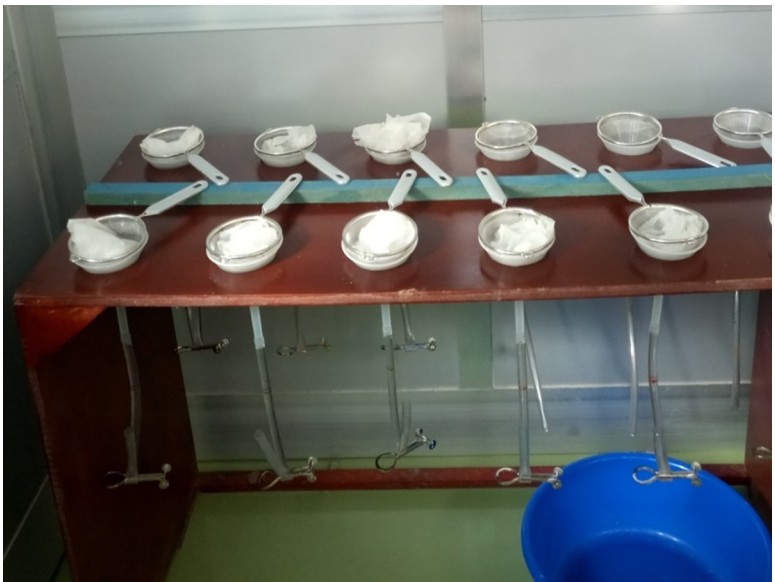

**Fig 2. Funnel stand with Baermann apparatus for filtering of *S. stercoralis* larvae from stool samples.**

**Stool collection and transportation.** On average, about 20 grams stool sample were collected from each participant using standard sterile stool collection cups without preservatives; all the samples were then labeled using unique identification numbers, placed in a triple packaging system and transported at room temperature to Bahir Dar University laboratory.

## Laboratory investigation procedures

**Conventional Baermann (CB).** Ten grams of stool sample were weighed and mixed with two grams of activated charcoal and lukewarm water. The stool sample was transferred to a petri dish with a double layer of tissue paper at the bottom and then covered by single layer of tissue paper at the top to form a small pouch. Incubation was maintained for 18–24 hrs at 26°C. After incubation, stool samples were suspended for 1 hour in lukewarm water at room temperature (which is in our case between 25 and 37°C) and filtered using a conventional Baermann apparatus (a strainer on top of a funnel connected to a rubber hose clamped with a hemostatic clamp) supported by a funnel stand, the single layer tissue paper side of the pouch was facing the strainer, (Fig 2). Afterwards, the lower 10 mL from the water contained in the hose was drained off, centrifuged at 2000 rpm for 5 minutes and 1 mL of the sediment was examined microscopically for the presence of larvae [6]. The detail procedure is given in S1 Text.

**Modified Baermann (MB)**

Three grams of fresh stool sample was weighed and placed on cotton-wool gauze (8 layers–non-sterile) of 5x5 cm. A stool pouch was formed and placed on top of 50 mL falcon tube filled with lukewarm water just slightly touching the water surface (Fig 3). The tube was left to stand at room temperature (which is in our case between 25 and 37°C) for 2.30 hours. The supernatant was discarded and the 3 mL sediment was allowed to settle for 30 minutes and then examined under the microscope for the presence of larvae [4]. The detail procedure is given in S2 Text.

**Modified Baermann with charcoal pre-incubation (MBCI)**

This is a newly modified version of MB technique. The materials and procedures used were the same as MB but with addition of charcoal pre-incubation. Briefly, three grams of fresh stool sample were weighed and mixed with one gram of activated charcoal and lukewarm water. Then a stool pouch, as in the regular MB, was formed using cotton-wool gauze (8

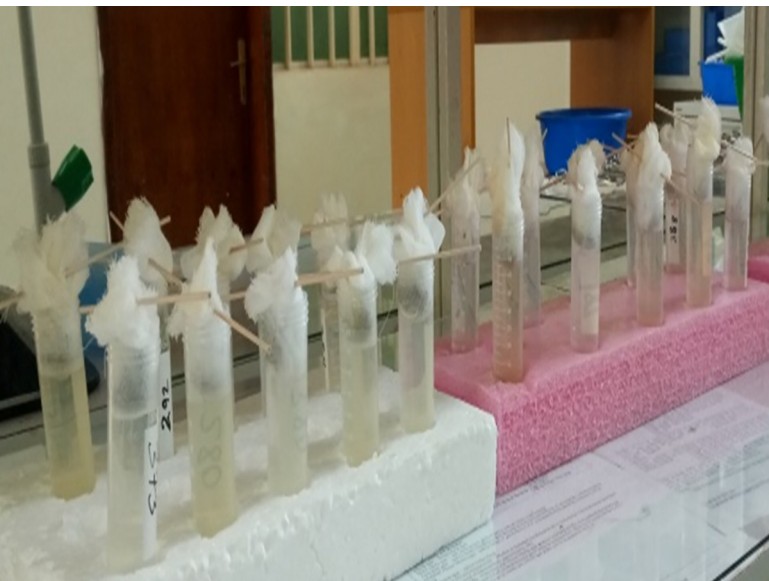

**Fig 3. Suspension of stool samples in 50 mL tubes with lukewarm water.**

layers–non-sterile) of 5x5 cm and placed in the center of a petri dish. Incubation was done for 18–24 hrs at 26˚C, and then processed as in the regular MB. The detail procedure is given in S3 Text.

### Reference standard test

Since there is no 'gold reference standard' for the Baermann technique, the results of all the three Baermann procedures were used in combination as a composite reference standard by which the presence of *S. stercoralis* larvae in at least one of the three Baermann procedures was used as a positive composite reference standard and the absence of *S. stercoralis* larvae by all the three Baermann procedures was used as a negative composite reference standard.

## Time and cost required to perform each Baermann technique

After the stool sample arrived at the laboratory, the actual hands-on time required to perform stool sample preparation was recorded. This was done by timing the average number of minutes spent on weighing the samples, incubating with activated charcoal, material preparation, filtration, centrifugation and microscopic examination for each Baermann technique. Incubation times and waiting times between steps were not included. In addition, the cost required to perform each Baermann technique was estimated by summing up the costs of the laboratory materials used. Costs were collected in Ethiopian Birr and converted to US dollars using an average of one year exchange rate from July, 2109 to June 2020, ranging from $28.4 to $34.3 (https://www.poundsterlinglive.com). The detailed cost estimation is given in S1 Table.

## Quality control

All laboratory procedures were performed by qualified technicians. Each stool sample was processed within 4 hours after collection. The laboratory procedures were done following the standard bench protocols prepared for the study and positive results were crosschecked by at least two experienced microscopists.

### Data management and statistical analysis

Laboratory results were recorded on paper forms by laboratory technicians and then entered to an excel sheet (Microsoft Excel, 2010) along with demographic data. Statistical analysis was performed on SPSS version 23 and openEpi software **Version 3.01 (**www.OpenEpi.com**, updated 2013/04/06).** Descriptive statistics was used to summarize the data and the diagnostic performance of the Baermann methods was assessed as sensitivity, specificity, Negative Predictive Value (NPV) and Positive Predictive Value (PPV) against the composite reference standard. McNemar test was used to compare the sensitivity, specificity and relative predictive values between the methods and Cohen's kappa was used to test the agreement of each method with the composite reference standard. P-values <0.05 were considered to be statistically significant.

## Results

Within 3 weeks of enrolment, 442 participants were found to be eligible. Of these, 5 were excluded because three participants gave too low quantity of stool samples at the time of collection and two participants gave enough quantity but were not fresh stool samples. Finally, samples from 437 participants were included and among these all three Baermann procedures were performed for 364 stool samples. Detailed distributions of samples analyzed in each procedure are presented in Fig 4.

### Socio demographic characteristics

Out of the 437 participants included, 268 (61.3%) were female. The age of the participants ranged from 2 to 76 years with a median age of 25 years. Overall, 151 (34.6%) individuals had detectable *S. stercoralis* larvae in at least one of the three Baermann techniques. The prevalence of strongyloidiasis varied between study villages, ranging from 9.6% in Gedro to 47.3% in

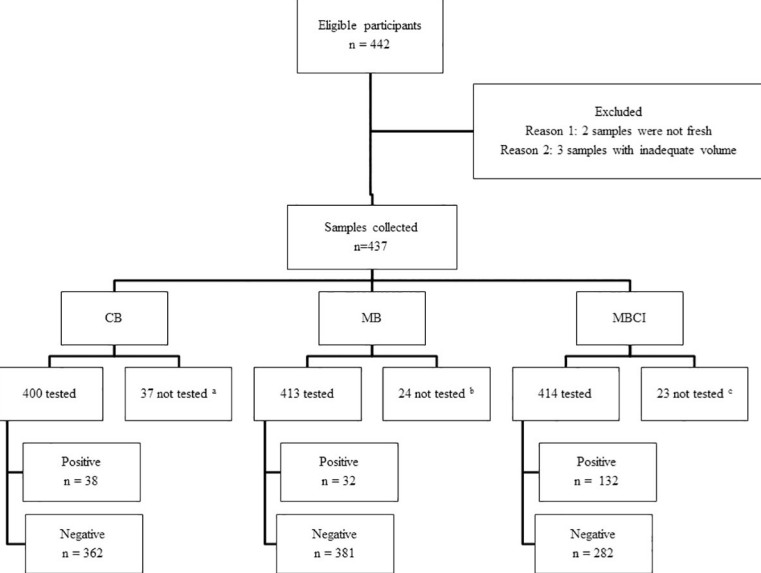

**Fig 4. Study participants' flow chart.** [a]Thirty-seven participants did not provide enough stool specimens to perform CB; [b]Twenty-four participants did not provide enough stool specimens to perform MB; [c]Twenty-three participants did not provide enough stool specimens to perform MBCI.

**Table 1. Strongyloidiasis Positivity and Negativity in relation to demographic characteristics.**

| Socio demographic characteristics | | Positive N (%) | Negative N (%) | TotalN (%) | AOR (95%CI) | P- Value |
|---|---|---|---|---|---|---|
| **Age** | <5 | 1 (20) | 4 (80) | 5 (1.1) | 2.64 (.237–29.38) | .429 |
| | 5–14 | 42 (30) | 98 (70) | 140 (32) | 2.96 (1.38–6.34) | .005 |
| | 15–49 | 87 (34.8) | 163 (65.2) | 250 (57.2) | 1.86 (.919–3.79) | .084 |
| | >49 | 21 (50) | 21 (50) | 42 (9.6) | 1 | - |
| **Sex** | Female | 80 (29.9) | 188 (70.1) | 268 (61.3) | 1.69 (1.09–2.58) | .017 |
| | Male | 71 (42) | 98 (58) | 169 (38.7) | 1 | - |
| **Residence** | Baynesa | 23 (45.1) | 28 (54.9) | 51 (11.7) | .241 (.058 - .998) | .050 |
| | Gedro | 5 (9.6) | 47 (90.4) | 52 (11.9) | 1.958 (.394–974) | .412 |
| | Gudguad | 53 (47.3) | 59 (52.7) | 112 (25.6) | .213 (.055 - .828) | .026 |
| | Kazimosh | 28 (40.6) | 41 (59.4) | 69 (15.8) | .292 (.072–1.178) | .084 |
| | Sesaberet -1 | 22 (33.3) | 44 (66.7) | 66 (15.1) | .398 (.099–1.598) | .616 |
| | Sesaberet -2 | 17 (25) | 51 (75) | 68 (15.6) | .694 (.167–2.88) | .616 |
| | Zenzelima | 3 (15.5) | 16 (84.2) | 19 (4.3) | 1 | - |

Gudguad. Strongyloidiasis positivity and negativity in relation to socio demographic characteristics is summarized in Table 1.

**Comparison between Baermann procedures.** Using CB, *S. stercoralis* larvae were detected in 38 of the 400 (9.5%) stool samples while using MB, 32 of the 413 stool samples (7.7%) were positive. The MBCI was positive for 132 of 414 (31.9%) of the samples. When limiting the analysis to the 364 samples examined by all three Baermann procedures; the number of positives by CB, MB and MBCI were 35 (9.6%), 29 (8.0%), and 114 (31.3%) respectively. Overall, 131 (36.0%) participants were found to be positive for *S. stercoralis*. This study also showed that the sensitivity of CB, MB and MBCI were 26.7%, 22.1% and 87% respectively. The performance and agreement of each method against the CRS is summarized in Table 2.

There were no significant differences between the sensitivities and NPVs of CB and MB (P-value = 0.33). On the other hand the sensitivity and NPV of MBCI was significantly higher than the sensitivities and NPVs of CB and MB (P-value = <0.001).

The time required to process each stool sample per technician by the three Baermann procedures following each step in the protocol is summarized in Table 3.

The total time (min) required to process each sample by CB, MB and MBCI was 16, 12, and 14 respectively without considering incubation and waiting time. The MB required less time to analyze each stool sample and results can be obtained on the same day of sample collection

**Table 2. Diagnostic performance of the three Baermann techniques against the composite reference standard.**

| | | Gold standard (Composite result) | | | Performance | | |
|---|---|---|---|---|---|---|---|
| | | Positive (%) | Negative (%) | Total (%) | Sensitivity(95% CI) | NPV(95% CI) | Kappa |
| | **Total** | 131 (100) | 233 (100) | 364 (100) | | | |
| CB | **Positive** | 35 (26.7) | 0 (0) | 35 (9.6) | 26.7 (19.9–34.9) | 70.8 (65.7–75.5) | 0.318 |
| | **Negative** | 96 (73.3) | 233 (100) | 329 (90.4) | | | |
| MB | **Positive** | 29 (22.1) | 0 (0) | 29 (8) | 22.1 (15.9–30.0) | 69.6 (64.4–76.3) | 0.267 |
| | **Negative** | 102 (77.9) | 233 (100) | 335(92) | | | |
| MBCI | **Positive** | 114(87) | 0 (0) | 114 (31.7) | 87 (80.2–91.7) | 93.2 (89.4–95.7) | 0.896 |
| | **Negative** | 17(13) | 233 (100) | 250 (68.7) | | | |

95% CI = 95% Confidence Interval, NPV = Negative predictive value

**Table 3. Time required to process three Baermann techniques.**

| Activities | | Required time per sample | | |
| --- | --- | --- | --- | --- |
| | | CB | MB | MBCI |
| Day 1 | Weighing (min) | 3 | 1 | 1 |
| | Mixing with activated charcoal and transfer to petri dish (min) | 2 | - | 2 |
| | Set up of materials for filtration (min) | - | 1 | - |
| | Filtration (hour:min) | - | 2:30 | - |
| | Collection of the fluid (min) | - | 2 | - |
| | Sedimentation (min) | - | 30 | - |
| | Microscopic Examination (min) | - | 8 | - |
| Day 2 | Set up of materials for filtration (min) | 3 | - | 1 |
| | Filtration (hour:min) | 1 | - | 2:30 |
| | Centrifugation (min) | 5 | - | - |
| | Collection of the fluid (min) | 5 | - | 2 |
| | Sedimentation (min) | - | - | 30 |
| | Microscopic Examination (min) | 3 | - | 8 |

CB—Conventional Baermann; MB—Modified Baermann; MBCI—Modified Baermann with charcoal pre-incubation

while CB and MBCI required two days (18–24 hr incubation) to get the result. Although the difference in the time required to analyze each sample by MBCI compared to CB seems small, 2 minutes per sample could have an impact especially in large scale screening initiatives by minimizing the overall time for testing. Thus, enables the technicians to handle more samples per person per day that will subsequently result in decreasing labor cost and maximizing the number of people who can be tested in a given time.

The cost effectiveness of the Baermann methods was estimated based on the consumable cost required per sample and the equipments used in each method with the assumption of 24 samples analyzed by CB and 40 samples by MB and MBCI methods daily. The total cost required to process each sample by CB, MB and MBCI was 15, 11, and 11.2 USD respectively and all the costs of each item needed in each procedure is presented in S1 Table.

## Discussion

*S. stercoralis* is among the most neglected NTDs, with a burden estimated in up to 400 million infected people worldwide [1,21], four times larger than previous estimates [22]. This difference in the global burden of *S. stercoralis* is due to limitations in diagnostic techniques that have led to a shortage of epidemiological data and its exclusion from the control strategies. A new scenario of raising awareness in the importance of *S. stercoralis* and the plans to establish a control program by 2030 implies that affordable and sensitive rapid diagnostic methods are needed to assess the true burden of prevalence and the amount of medication needed for deworming programs.

A previous study showed that among the microscopy-based techniques, sedimentation concentration had better sensitivity of 88% in recovering the *S. stercoralis* larvae from stool samples followed by Baermann (81%), agar plate culture (58%), and Harada-Mori (50%) [23]. Another study showed that the diagnostic agreement of Baermann funnel and Koga agar plate techniques with PCR was highest when the *S. stercoralis* infection intensity is high [24] although the later showed decreased detection rate when the infection intensity is low. These differences reported by different research groups for the same diagnostic technique are mainly due to the lack of harmonization of the technique.

In this study the overall prevalence of *S. stercoralis* infection was 34.6% which is comparable to results from a previous study using CB in this area of Ethiopia [6]. Although prevalence of *S. stercoralis* infection increases with age [9], in our study age [5–14] was found to be significantly associated with the infection, P–value = 0.005. This might be due to the lack of shoe wearing habit of the children in the study area that might increase their exposure to the infection. Similarly, sex (Female) and residence (Being resident of Gudguad) found to be statistically significant with P–value = <0.05. This might be due to most of the participants were females (61.3%) and residents of Gudguad village. Since *S. stercoralis* is not incorporated in the national STH control strategies and no specific interventions have been implemented, the prevalence remains the same over the years. Here we present, to our knowledge, the first comparison of different Baermann techniques for the diagnosis of *S. stercoralis* infections. Our results show that the modifications of the CB technique resulted in significant differences in the sensitivity and therefore in the calculation of the prevalence. In relation to this, it is observed that the sensitivity of CB did not show significant difference when compared to the sensitivity of MB. The significantly higher sensitivity achieved by MBCI in the diagnostic performance when compared to the MB (P–value = <0.001) is probably related to the incubation procedure with activated charcoal, which allows the parasite to enter the free living cycle and increases its detection. Morphological identifications were used to differentiate the stages of *S. stercoralis* larvae from hookworm larvae. We have confirmed that all the *S. stercoralis* larvae detected by MB were rhabditiform larvae while both rhabditiform and filariform larvae were detected by MBCI and CB. On the other hand, the performance of MBCI was significantly superior to the CB (P–value = <0.001). MBCI demonstrated sensitivity three times higher than CB, which might be due to the application of the gauze used in the filtration step in the case of MBCI that has more permeability to the larvae to migrate towards the warm water than the tissue paper that we used for CB. Previous studies used either tissue papers or gauze for filtration [4,6,24] but so far there is no standard tissue paper or gauze recommended for the filtration step either in CB or MB. For this study, locally available tissue roll paper was used for the CB and gauze for MB and MBCI which is adopted in the MB procedure from the Leiden University Medical Center's (LUMC) in the Netherlands [4]. In addition, the reduced sample volume in MBCI might enable the larvae to move freely and migrate towards the warm water. The stool samples were processed by the three Baermann techniques simultaneously using the same type of water, water temperature, and in same laboratory set up with similar lighting conditions.

Among the microscopy-based techniques, the CB is the Baermann technique most commonly used for the routine diagnosis of *S. stercoralis* infection. In this study we are providing evidence that critical steps in the Baermann technique like the charcoal pre-incubation and the fabric used for larval migration (tissue paper vs gauze) significantly affect its sensitivity. Our data shows that the MBCI is easy to implement, simple to perform, does not require a centrifugation process and funnels, comparatively cheaper and requires much less space in the laboratory than the CB. In addition, MBCI requires less time to get results when compared to coproculture and Koga nutrient agar plates [4,25]. This enables the technique to be adapted and is easily accessible for field-based epidemiological studies and clinical trials especially in large scale studies. Besides its advantages in sensitivity, it is also relevant that the high NPV achieved by the MBCI, reaching 93.2% in this evaluation, might make it a useful method in the assessment of treatment response in clinical trials. The Baermann technique for the assessment of treatment response is challenging since lack of detection of larvae cannot always be indicative of cure. This is because of the individual patterns of larval shedding in stool. This limitation can be overcome by performing consecutive stool samples at screening and follow-up post treatment. The added benefit of the MBCI is that since it is simple to perform it can be

employed for consecutive day screening. Although serology has shown a good performance as a test of cure in non-endemic settings [22], that tool is of little value in endemic areas where re-infection is likely.

Another drawback of the Baermann method is that, if stool samples contain hookworm eggs, these may hatch during the Baermann procedure. However, this limitation can be overcome by distinguishing the morphology of the different species.

The fact we used a single stool sample might, despite an optimal Baermann procedure, still give an underestimation of the actual prevalence of *S. stercoralis* infection in our study population. Alternative techniques with a higher sensitivity, such as Koga agar plate culture or PCR, could not be used due to the lack of appropriate laboratory facilities. In general, techniques based on the detection of parasite DNA remain a challenge in many regions where strongyloidiasis is endemic.

In conclusion, our evaluation shows that the modified Baermann with charcoal pre-incubation is a sensitive and affordable diagnostic approach for microscopic detection of *S. stercoralis* larvae in stool. It is comparatively cheaper, uses less laboratory space and is relatively simple to implement, therefore offering a practical tool for the diagnosis of *S. stercoralis* infections in surveys and clinical trials.

## Supporting information

**S1 STARD-2015-Checklist.**
(DOCX)

**S1 Table. Template for cost estimation.**
(DOCX)

**S1 Text. SOP for Conventional Baermann.**
(DOCX)

**S2 Text. SOP for Modified Baermann.**
(DOCX)

**S3 Text. SOP for Modified Baermann with charcoal pre-incubation.**
(DOCX)

## Acknowledgments

The STOP project consortium includes: Jose Munoz, Lisette van Lieshout, Alejandro J. Krolewiecki, Charles Mwandawiro, Rachel Pullan, Inacio Mandomando, Maria Martinez Valladares, Wendemagegn Enbiale Yeshaneh, Rafael Guille, Nana Aba Williams, Rafael Balana Fouce, Marc Fernandez, Adelaida Sarukhan, Helena Marti-Soler, Berta Grau Pujol, Javier Gandasegui, Valdemiro Escola, Stella Kepha, Martin Rono, Ellie Baptista, Graham Medley, Catherine Pitt, Augusto Messa Junior, Maria Cambra Pelleja, Maria Demontis, and Woyneshet Gelaye. The authors would like to acknowledge study participants for their participation in the study, Zenzelima health center staff, the health extension workers and the laboratory staff members of Bahir Dar University, special thanks goes to Hiwot Tadesse.

## Author Contributions

**Conceptualization:** Woyneshet Gelaye, Nana Aba Williams, Stella Kepha, Augusto Messa Junior, Helena Marti-Soler, Alejandro J. Krolewiecki, Lisette van Lieshout, Wendemagegn Enbiale.

**Data curation:** Woyneshet Gelaye.

**Formal analysis:** Woyneshet Gelaye, Pedro Emanuel Fleitas, Helena Marti-Soler.

**Investigation:** Woyneshet Gelaye, Stella Kepha, Augusto Messa Junior, Pedro Emanuel Fleitas.

**Methodology:** Woyneshet Gelaye, Nana Aba Williams, Stella Kepha, Augusto Messa Junior, Pedro Emanuel Fleitas, Helena Marti-Soler, Alejandro J. Krolewiecki, Lisette van Lieshout, Wendemagegn Enbiale.

**Supervision:** Woyneshet Gelaye, Alejandro J. Krolewiecki, Lisette van Lieshout, Wendemagegn Enbiale.

**Validation:** Woyneshet Gelaye, Wendemagegn Enbiale.

**Writing – original draft:** Woyneshet Gelaye.

**Writing – review & editing:** Woyneshet Gelaye, Nana Aba Williams, Stella Kepha, Augusto Messa Junior, Pedro Emanuel Fleitas, Helena Marti-Soler, Destaw Damtie, Sissay Menkir, Alejandro J. Krolewiecki, Lisette van Lieshout, Wendemagegn Enbiale.

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
