## [Decision Letter · Decision Letter 0]

18 Sep 2020

Dear Gelaye,

Thank you very much for submitting your manuscript "Performance evaluation of Baermann techniques: the quest for developing a microscopy reference standard for the diagnosis of Strongyloides stercoralis" for consideration at PLOS Neglected Tropical Diseases. As with all papers reviewed by the journal, your manuscript was reviewed by members of the editorial board and by several independent reviewers. In light of the reviews (below this email), we would like to invite the resubmission of a significantly-revised version that takes into account the reviewers' comments. 

Please note the attached file by reviewer 3.

We cannot make any decision about publication until we have seen the revised manuscript and your response to the reviewers' comments. Your revised manuscript is also likely to be sent to reviewers for further evaluation.

Sincerely,

Peter Steinmann, Ph.D.

Associate Editor

Ricardo Fujiwara

Deputy Editor

Reviewer's Responses to Questions

**Key Review Criteria Required for Acceptance?**

**Methods**

-Are the objectives of the study clearly articulated with a clear testable hypothesis stated?

-Is the study design appropriate to address the stated objectives?

-Is the population clearly described and appropriate for the hypothesis being tested?

-Is the sample size sufficient to ensure adequate power to address the hypothesis being tested?

-Were correct statistical analysis used to support conclusions?

-Are there concerns about ethical or regulatory requirements being met?

Reviewer #1: While advances are being made in the molecular diagnosis of Strongyloides stercoralis, this report makes is clear there is a need for an accurate and cheap standardized method for use in poor endemic areas. The authors therefore tested 3 modifications of the Baermann technique that are now in use to see which produced the most accurate diagnosis (the presence or absence of a living stage of S. stercoralis in the tested feces). The number of samples tested by each method were close to the calculated minimum needed. The statistics used were appropriate for this type of study. Ethical approval and regulatory approval were obtained before running the study.

Questions for the authors:

1. Line 98: Do you mean the Baermann technique is usually reserved (“mostly”) for field studies and clinical trials, or that the technique is the most often technique used in field studies and clinical trials? Please re-word this sentence to make your meaning clear. Where does it stand among techniques currently used for clinical diagnosis?

2. Lines 103-104: Why did you pick these three methods of running a Baermann technique for S. stercoralis, as I am sure that there are many other modifications used in diagnostic labs? Do you have any data on what proportion of clinical diagnostic labs are using each of the 3 methods you compare?

3. Lines 126-130: You calculated using Buderer’s formula that you needed a minimum of 432 samples for your study, but you only used 400 - 414 for any one test (364 by all 3 techniques). Why didn’t you collect enough samples to make your calculated minimum for each test run?

Reviewer #2: Please see general comments below.

Reviewer #3: -The objectives of the study are clear

-Is the study design is appropriate and the sample size sufficient statistical analysis used are correct

methodology descriptin should be revised and clarified

**Results**

-Does the analysis presented match the analysis plan?

-Are the results clearly and completely presented?

-Are the figures (Tables, Images) of sufficient quality for clarity?

Reviewer #1: The tables and figures are clear and the results are clearly presented. The study's analysis follows the plan to compare the 3 techniques on samples from the same study population. However the data produced by the study could have been analyzed further allowing further comparisons to published data. 

Questions for the authors:

4. Line 209-214: You give a breakdown of the sex and age of your participants, but you don’t bother to test if infections with S. stercoralis is significantly different in any subgroup or village. You should statistically test the results in each subgroup and report the significance if any. You could also let the reader know if any of the 3 techniques found significant number of infections within a subgroup while the other 2 did not.

5. Line 215: The MBCI modification most likely would have worked even better with more feces, was the size of the 50 ml tube’s opening the limiting factor (i.e. you used less feces to limit the size of the fecal pouch to that which would fit into the tube)?

6. Line 243: Why did you make the assumption in determining the cost that you would compare 24 CB tests daily to 40 of the other 2 tests daily?

7. Lines 264-265: Your study found 34.6% infection rate (3 combined tests) compared to reference #6 rate of 33% by Baermann (method not described in this meeting abstract). The overall rate of infection found in reference #6 was 56% using the Baermann and 2 other tests. Do you know which of your 3 methods most closely corresponds to the Baermann method used in reference #6?

Reviewer #2: Please see general comments below.

Reviewer #3: Table 2 is not usefull

**Conclusions**

-Are the conclusions supported by the data presented?

-Are the limitations of analysis clearly described?

-Do the authors discuss how these data can be helpful to advance our understanding of the topic under study?

-Is public health relevance addressed?

Reviewer #1: The necessity of a cheap but accurate microscopy technique for the diagnosis of S. stercoralis in endemic areas (very often within very poor countries) is made clear in the introduction of this study. Without such a test the goal of implementing a strategy to eliminate S. stercoralis can not begin. The authors' results also make it clear that one of the 3 techniques tested is significantly better that either or both of the other 2 and the cost is about the same as either of the inferior techniques.

Reviewer #2: Please see general comments below.

Reviewer #3: Conclusions are oversized and should be moderated

**Editorial and Data Presentation Modifications?**

Reviewer #1: Line 44: “that” should be “the”.

Reviewer #2: (No Response)

Reviewer #3: (No Response)

**Summary and General Comments**

Reviewer #1: This study concentrates on the cheap, accurate and widely used Baermann technique for the diagnosis of S. stercoralis while most recent publications compare expensive and hard to preform molecular techniques to microscopy techniques that are currently in place. Until the third world catches up to the standards needed to run PCR diagnostics (and recent events with CoVid-19 suggest even the developed world has a way to go!) only tests with few "moving parts" are going to be useful. As the authors state there must be a "gold standard" Baermann test for the diagnosis of S. stercoralis before we can move on to eliminating its threat. The authors could have used the Discussion to suggest the steps needed to run a gold standard Baermann.

Reviewer #2: This manuscript reports on different variations of the Baermann technique for the microscopic diagnosis of Strongyloides stercoralis. Strongyloidiasis is increasingly being recognized as one of the clinically most important helminth infections, and standard diagnostic tools are insufficiently sensitive. The considerably varying reported diagnostic accuracies for the different detection techniques (Baermann, nutrient agar plate, PCR) might partially be explained by a lack of standardization of these techniques. Hence, the current manuscript is of importance and I support documentation of this work in published form.

Specific comments:

- Introduction & Methods: It would be helpful to provide a distinct reference for each of the three Baermann variations.

- Methods: For the sample size calculation, you estimated a 32% prevalence, which is close to the actual prevalence found in the study. Did you have preliminary data on the occurrence of S. stercoralis in the study area or what assumption was this estimate based on?

- Results: Please detail why five participants were excluded „due to issues with the samples“ (too low quantity?).

- Results & Discussion: The authors show considerable differences in the prevalence obtained by the three modifications of the Baermann technique. Did they develop specific SOPs for the different protocols and could these be shared, e.g. as supplementary files? Alternatively, a schematic figure illustrating the three test modifications would be really useful. Additionally, it would be helpful to discuss e.g. the effects and mechanisms of charcoal incubation and the different gauze layers more thoroughly. What about other factors, e.g. type of water used, water temperature and the role of light, which is also commonly used to facilitate excretion of S. stercoralis larvae from the pouch to the Baermann tube?

- Discussion: Some of the Baermann modifications lead to a prolonged incubation so that it takes 24 hours or more until results become available. Hence, when these modifications are used, there is not much time difference as compared to charcoal culture or Koga nutrient agar plates. This could be mentioned/discussed.

- Discussion: I miss a part on limitations of this study. With all the detailed efforts made to accurately compare the different Baermann tests, would it not have been a good opportunity to perform also PCR and or Koga agar plate, in particular to obtain a more independent composite reference standard?

Reviewer #3: study of some interest in the field work with an important background in the main issue, which is the appropriate management of STH and the inclusin of s. stercorlais in the NTD.

PLOS authors have the option to publish the peer review history of their article (what does this mean?). If published, this will include your full peer review and any attached files.

Reviewer #1: No

Reviewer #2: Yes: Sören L. Becker

Reviewer #3: No
---

## [Editor Report · Decision Letter 1]

15 Dec 2020

Dear Gelaye,

We are pleased to inform you that your manuscript 'Performance evaluation of Baermann techniques: the quest for developing a microscopy reference standard for the diagnosis of Strongyloides stercoralis' has been provisionally accepted for publication in PLOS Neglected Tropical Diseases.

Best regards,

Peter Steinmann, Ph.D.

Associate Editor

Ricardo Fujiwara

Deputy Editor

---

## [Editor Report · Acceptance letter]

28 Jan 2021

Dear Gelaye,

We are delighted to inform you that your manuscript, "Performance evaluation of Baermann techniques: the quest for developing a microscopy reference standard for the diagnosis of *Strongyloides stercoralis*," has been formally accepted for publication in PLOS Neglected Tropical Diseases.

Best regards,

Shaden Kamhawi

co-Editor-in-Chief

Paul Brindley

co-Editor-in-Chief
